# The impact of city financial ecology on firm financing efficiency: Evidence from China's strategic emerging industries

**Hanbo Zhang** [1]⊙, **Guiyang Zhang** [1]⊙, **Yong Qi** [2]*, **Yuchen Gao** [3], **Dong Wang** [4], **Xing Li** [1]

1 School of Economics & Management, Nanjing University of Science & Technology, Nanjing, Jiangsu, China, 2 School of Intellectual Property, Nanjing University of Science & Technology, Nanjing, Jiangsu, China, 3 School of Public Policy & Management, Tsinghua University, Beijing, China, 4 School of Public Administration, Nanjing University of Finance & Economics, Nanjing, Jiangsu, China

⊙ These authors contributed equally to this work.
* qyong@njust.edu.cn

**Data Availability Statement:** All relevant data are within the paper and its Supporting information files.

## Abstract

Based on the concept of bionics and the connotation of city financial ecology, this study constructs a 3-level and 27-indicator evaluation index system, including financial ecology growth, soil, and air. This study uses the entropy-TOPSIS model to weigh indicators objectively and evaluate the financial ecology of 343 China's prefecture-level cities during 2009–2016. This study uses the DEA-Tobit method to assess the financing efficiency of 4013 China's strategic emerging listed firms during 2010–2017 and runs random-effect Tobit panel regressions. Regression results suggest that a city's financial ecology overall has a positive effect on strategic emerging firms' financing efficiency. Therefore, the government should: improve the multi-tiered financial market system and encourage financial innovation; transform the economic growth model and optimize the industrial structure; establish an information-sharing mechanism and construct a social credit system.

## Introduction

Since financing theory and efficiency theory emerged in the 1950s, firm financing efficiency has been increasingly investigated [1, 2]. However, most research focuses on startups and SEMs [3, 4], ignoring strategic emerging industries. With the release of "The Decision of the State Council on Accelerating Fostering and Development of Strategic Emerging Industries" in 2010, China has upgraded strategic emerging industries to the national strategic level. The capital is the core artery for strategic emerging industries' development. According to the survey report issued by the Ministry of science and technology of China in 2013, more than 55% of R&D projects in strategic emerging areas are not further implemented due to insufficient capital investment. There is an urgent need to improve the financing efficiency of China's strategic emerging industries.

　　Scholars have noticed the financing constraints in China's strategic emerging industries and assessed the financing efficiency with DEA, fuzzy evaluation, the grey relational degree, etc. For instance, some scholars have analyzed the financing efficiency of specific industries,

**Funding:** YQ received funding from the National Natural Science Fund of China (71974096). YG received funding from the National Natural Science Fund of China (72104121). GZ received funding from the National Social Science Fund of China (21CGL004, 21AZD012), the Philosophy and Social Science Fund of Jiangsu Province (20GLC012), the Fundamental Research Funds for the Central Universities (30921012201), the Jiangsu Province's 14th Five-year Educational Science Planning Project (C-b/2021/01/25), and the Young Teachers Research Foundation Project of School of Economics and Management in Nanjing University of Science and Technology (JGQN2002). The funders had no role in study design, data collection and analysis, decision to publish, or preparation of the manuscript.

**Competing interests:** The authors have declared that no competing interests exist.

such as high-end equipment manufacturing [5], new energy vehicles [6], new energy [7], and environmental protection [8]. Also, other scholars have investigated that the financing efficiency of strategic emerging industries in different regions, including eastern, central, and western China [9]. Their findings indicate that the financing efficiency of strategic emerging industries is not optimistic [10]. However, few scholars investigate its influencing factors or find ways to improve them.

Research in traditional industrial contexts suggests two influencing factors: internal fundamentals and external environment. The former includes credit status, corporate governance, capital utilization, profitability, etc. The latter contains financial market development, macroeconomic development, policy support, social credit environment, etc. Previous research emphasizes the internal fundamentals and empirically confirms their influences [3, 4, 10]. However, internal fundamentals could only explain the differences among specific firms rather than the systematic financing constraints of the whole strategic emerging industries [11].

Regarding strategic emerging industries, the external environment may outweigh the internal fundamentals. Strategic emerging firms are mostly in the seed stage, characterized by high uncertainty and insufficient collateral. In addition, their collateral is mainly in the form of intellectual property, which requires a well-developed intellectual property pledge, evaluation and trading system. In China, the juvenile system may constrain strategic emerging firms' use of intellectual property for pledge loans, transfers, or investment. Therefore, strategic emerging industrial firms could not rely on their internal fundamentals to obtain external funds.

With the introduction of "enterprise ecosystem" and "financial ecology" concepts, more and more scholars have begun to investigate the role of external financial ecology. However, most of them focus on one or two aspects. Only a few try to construct a comprehensive index. For instance, Li and Kuhn [12] propose that regional financial ecology contains nine factors: judiciary environment, government management, credit basis, etc. Xiong and Geng [13] select three significant factors: economic development, financial development, and honesty. Xu et al. [11] propose an index system that includes the macro-economy, the government's role, financial development, and credit environment. There is no consensus on the index system for financial ecology.

The major contributions of this this study are as follows: (1) Reclassifies the city's financial ecological into three categories (growth, soil, and air) on the ground of theoretical analysis from the perspective of bionics, further advancing the systematization of external financial ecological evaluation. Then, this study uses the entropy-TOPSIS model to assess the financial ecology of 343 China's prefecture-level cities from 2009 to 2016. (2) Fills the research gap on the relationship between city financial ecology and firm financing efficiency based on the DEA-Tobit method. This study uses 4013 strategic emerging listed companies in China from 2010 to 2017 as a sample, calculates their financing efficiency through the DEA method, and reveals the impact of city financial ecology on firm financing efficiency through the random-effect Tobit panel regression model.

## Theoretical analysis

Despite a lack of directly-related research on the nexus between city financial ecology and firm financing efficiency, scholars have confirmed the significance of an excellent financial environment in enhancing firm financing efficiency since very early [14]. This study proposes that the three perspectives of city financial ecology may positively affect firm financing efficiency.

First, financial ecology growth, referring to the status quo of the financial market, directly affects firm financing behavior. A well-developed financial market in terms of banks, securities, and residents provides strategic emerging firms with diversified financing ways [9, 15].

Among them, financial intermediaries, represented by banks, are capable of passing the funds from surplus sectors towards deficient sectors to augment credit supply, thereby ultimately propelling economic growth based on the supply-leading theory [16]; the stock market is a fundamental financial part of a country's economy, which channelizes funds, connects savers to investors, and enables listed firms to go for technological development, leading to economic growth ultimately [17]. In addition, evidence suggests that direct financing sources help loosen the financial constraint that strategic emerging firms face and thus ease financing dilemmas [18].

Second, financial ecology soil, referring to the direct material basis of financial ecology, provides economic and policy support for financial ecology growth. A solid macro-economic foundation means adequate market capital supply, reducing the financing difficulty and improving financing efficiency [19]. Governments can provide direct fiscal subsidies by formulating the financial scheme [20]. Such subsidies accelerate the development of new technology [21], which is the core collateral for financing. Governments also help strategic emerging firms get legitimacy and receive investments from markets [22].

At last, financial ecology air, referring to the external environment that may affect investors' perception of returns and risks, may also affect financing efficiency. For instance, credit is the basis of lending. A good credit environment can reduce the information asymmetry between the demand the supply of funds. It helps build mutual trust, reduce transaction costs, and improve financing efficiency [11]. Other institutional environments, e.g., human capital, education, and security environments, may favor financing efficiency by enhancing investors' confidence in returns and reducing their perceptions of risks [23].

## Methodology

### Index design

**Bionics-based index of city financial ecology.** Based on the concept of bionics and the connotation of financial ecology, this study develops a bionics-based index of city financial ecology. Table 1 shows the details.

This study calculates Cronbach's alpha with Z-score standardized data to test the reliability. As shown in Table 2, all the Cronbach's α is ≥0.789, suggesting a good consistency, stability, and reliability.

**Input-output index of firm financing efficiency.** Following previous DEA practice, this study selects four input and output indexes, respectively. Table 3 shows the details.

### Entropy-TOPSIS model

The entropy-TOPSIS model is widely accepted and matching with the economic question addressed in this study, for the following reasons: (1) The entropy method is an objective weighting method as opposed to subjective weighting methods like the Analytic Hierarchy Process (AHP), which can both lessen the subjectivity of indexes' weight and intuitively judge the effectiveness of indexes' information content, more in line with the demands of practical operations. (2) The Technique for Order Preference by Similarity to an Ideal Solution (TOPSIS) method has the benefits of straightforward computation, modest sample size need, and obvious and accurate results. The advantages of the previous two methods are combined in the entropy-TOPSIS model, which contributes to more objective and logical evaluation results. Therefore, this study uses the entropy-TOPSIS model to evaluate the city's financial ecology.

**Table 1. The index of city financial ecology.**

| Level 1 | Level 2 | Level 3 |
|---|---|---|
| Financial ecology growth | Financial institution | Deposit balance |
| | | Loan balance |
| | | Financial deepening: (deposits + loans) / GDP |
| | | Financial utilization: loan balance / deposit balance |
| | Enterprise | Gross industrial value of enterprises above designated size |
| | | Proportion of tertiary industry production value |
| | Resident | Resident savings deposit |
| | | Per capita consumer spending |
| Financial ecology soil | Economic base | Per capita GDP |
| | | Growth rate of GDP |
| | | Growth rate of investment in fixed assets |
| | Policy base | Proportion of fiscal revenue |
| | | Proportion of fiscal expenditure |
| | | Proportion of foreign direct investment |
| Financial ecology soil | Credit | Coverage of credit system |
| | | Proportion of dishonest market entities |
| | Human resource & education | Number of college students |
| | | Number of stuffs |
| | Public security | Proportion of staff attending endowment insurance |
| | | Proportion of staff participating in unemployment insurance |

The steps for using the entropy weight TOPSIS model are as follows [24, 25]: First, we construct an initial score matrix as follows:

$$X = \left[x_{ij}\right]_{m \times n} \tag{1}$$

where $x_{ij}$ indicates the score of city $i$ in terms of indicator $j$, $m$ and $n$ are the numbers of cities and indicators, respectively.

Second, we perform the processing dimensionless to eliminate the influence of index dimension and obtain dimensionless score matrix $X^*$ through the following transformation:

$$x_{ij}^* = \frac{x_{ij} - \min x_{\cdot j}}{\max x_{\cdot j} - \min x_{\cdot j}} \tag{2}$$

for positive indicators, and:

$$x_{ij}^* = \frac{\max x_{\cdot j} - x_{ij}}{\max x_{\cdot j} - \min x_{\cdot j}} \tag{3}$$

for negative indicators.

**Table 2. Cronbach's alpha for financial ecology index.**

| Year | 2009 | 2010 | 2011 | 2012 | 2013 | 2014 | 2015 | 2016 |
|---|---|---|---|---|---|---|---|---|
| Cronbach's α | 0.865 | 0.789 | 0.857 | 0.846 | 0.863 | 0.841 | 0.866 | 0.858 |

**Table 3. Inputs and outputs for firm financing efficiency.**

| Input/output | Indicators | Definition |
|---|---|---|
| Input | Bank credit financing | (long term loans + short term loans) / total assets |
| | Equity financing | (equity + capital reserves) / total assets |
| | Internal financing | (surplus reserves + undistributed profits) / total assets |
| | Government subsidy | government subsidies / total assets |
| Output | Return on net assets | after-tax profits / net assets |
| | Return on capital | after-tax profits / paid-in capital |
| | Growth of total operating income | (operating income at the end of the current period—operating income at the end of the last period) / operating income at the end of the last period |
| | Earnings per share | after-tax profits / equity |

Next, we obtain the weighted score matrix as follows:

$$Y = \left[ y_{ij} \right]_{m \times n} = \left[ x_{ij} \times w_j \right]_{m \times n} \tag{4}$$

where $w_j$ refers to the entropy-based weighted coefficient of indicator $j$:

$$\begin{cases} w_j = \left( 1 - e_j \right) \Big/ \sum_{j=1}^{n} \left( 1 - e_j \right) \\[2ex] e_j = - \sum_{i=1}^{m} p_{ij} \ln p_{ij} \\[2ex] p_{ij} = x_{ij} \Big/ \sum_{i=1}^{m} x_{ij} \end{cases} \tag{5}$$

After the entropy-based weighted normalization process, we obtain each the positive ideal solution and negative ideal solution of each indictor, in which:

$$\begin{cases} y_j^+ = \max_{1 \le i \le m} y_{ij} \\[2ex] y_j^- = \min_{1 \le i \le m} y_{ij} \end{cases} \tag{6}$$

and calculate the Euclidean distance of city $i$ to positive ideal solution and negative ideal solution, in which:

$$\begin{cases} s_j^+ = \sqrt{\sum_{i=1}^{n} \left( y_j^+ - y_{ij} \right)^2} \\[3ex] s_j^- = \sqrt{\sum_{i=1}^{n} \left( y_j^- - y_{ij} \right)^2} \end{cases} \tag{7}$$

**Table 4. Score distribution of city financial ecology.**

| Score | Number | Cities |
|---|---|---|
| [0.45, 1.00) | 2 | Beijing, and Shanghai. |
| [0.40, 0.45) | 2 | Shenzhen, and Tianjin. |
| [0.35, 0.40) | 8 | Guangzhou, Hangzhou, Ordos, Chongqing, Chengdu, Kunming, Haikou, and Sanya. |
| [0.30, 0.35) | 32 | Suzhou, Nanjing, Taiyuan, Guiyang, Xi'an, Xiamen, Xining, Wuhan, Lhasa, Dalian, etc. |
| [0.25, 0.30) | 127 | Zhuhai, Anshun, Dongying, Wenzhou, Guyuan, Wuzhong, Changchun, Baotou, Xinzhou, Zhangjiajie, etc. |
| (0.00, 0.25) | 115 | Heyuan, Yichun, Luoyang, Congzuo, Zibo, Handan, Guiyang, Chaoyang, Qiqihar, Dandong, etc. |

At last, we calculate each city's financial ecology score by calculating its closeness degree to the ideal solution as follows:

$$c_i = \frac{s_i^-}{s_i^+ + s_i^-} \tag{8}$$

The larger the $c_i$, the better financial ecology of city $i$, and vice versa.

Table 4 shows the score distribution of city financial ecology (average from 2009 to 2016). Beijing, Shanghai, Shenzhen, Tianjin and Guangzhou rank top five of all cities. Eastern China is superior to Central China and Western China. Coastal cities, especially those in the regions of Beijing-Tianjin, the Yangtze River Delta, and the Pearl River Delta, are better than others.

## DEA-Tobit method

The DEA-Tobit method is perfectly matching with the economic question addressed in this study, for the following reasons: (1) As a systematic method for efficiency evaluation, the Data Envelopment Analysis (DEA) method is being applied to multi input and multi output situations, and does not require dimensionless processing of data or weight coefficients, thereby avoiding the impact of subjective evaluation. (2) The efficiency value calculated by DEA ranges from 0 to 1, implying that independent variables can only be observed in a limited way in the context of research on influencing factors of efficiency, and using general regression models based on Ordinary Least Squares (OLS) may cause problems such as parameter estimation bias and inconsistency. As a typical censored regression model, the Tobit model can not only effectively solve the above problems, but also determine the direction and intensity of the impact on efficiency value according to the coefficients of influencing factors. Therefore, this study uses the DEA-Tobit method to assess financial efficiency and analyze its impact.

**DEA method.** This study evaluates the financing efficiency of 4013 list firms during 2010–2017 with the DEA method [26, 27] based on the following formula:

$$s.t. \begin{cases} \min \theta \\ \sum_{k=1}^{n} \lambda_k X_{ik} + s^- = \theta X_t \\ \sum_{k=1}^{n} \lambda_k Y_{jk} - s^+ = Y_t \\ 1 \leq t \leq n \end{cases} \tag{9}$$

where $\theta \in (0,1]$ refers to the efficiency, $n$ refers to the number of decision-making unit (DMU), $i$ and $j$ refer to the number of input and output of each DMU, $\lambda_k \geq 0$ refers to the weight of DMU $k$, $X_{ik}$ refers to the input value $i$ of DMU $k$, $Y_{jk}$ refers to the output value $j$ of DMU $k$, $s^- \geq 0$ and $s^+ \geq 0$ refer to the slack variable of input and output. If $\theta = 1$ and $s^- = s^+ = 0$, it can be determined that the DMU has the optimal efficiency with a constant return-to-scale.

Fig 1 shows the financing efficiency of each strategic emerging industry annually. As shown, the financing efficiency of all strategic emerging industries shows a fluctuating downward trend.

**Random-effect Tobit panel regression model.** This study controls firm-level fundamentals, industrial dummies, regional dummies, and market board dummies. Table 5 shows the details.

The value of the dependent variable ranges from 0 to 1, requiring a Tobit regression model. The sample consists of 4013 firms spanning eight years, requiring a panel regression model. The coefficients' variance is systematic at the firm dimension according to Hausman test [28], requiring a random-effect regression mode. Taken together, we adopt the random-effect Tobit panel regression model as follows:

$$
\begin{aligned}
Fineffi_{i,t} &= \beta_0 + \beta_1 Finecol_{i,t-1} + \beta_2 TA_{i,t} + \beta_3 ATR_{i,t} + \beta_4 ALR_{i,t} + \beta_5 FCFPS_{i,t} \\
&\quad + \beta_6 Age_{i,t} + Indt + Region_l + Board_m + \varepsilon_{i,t}
\end{aligned}
\tag{10}
$$

where $i = 1,2,\cdots,4013$, $t = 1,2,\cdots,8$, $k = 1,2,\cdots,7$, $l = 1,2$, $m = 1,2,3,4$.

## Results and discussion

### Data and descriptive statistics

We set this study in China's strategic emerging industries. The official document "13th Five-Year Development Plan for Strategic Emerging Industries" has identified 7 strategic emerging

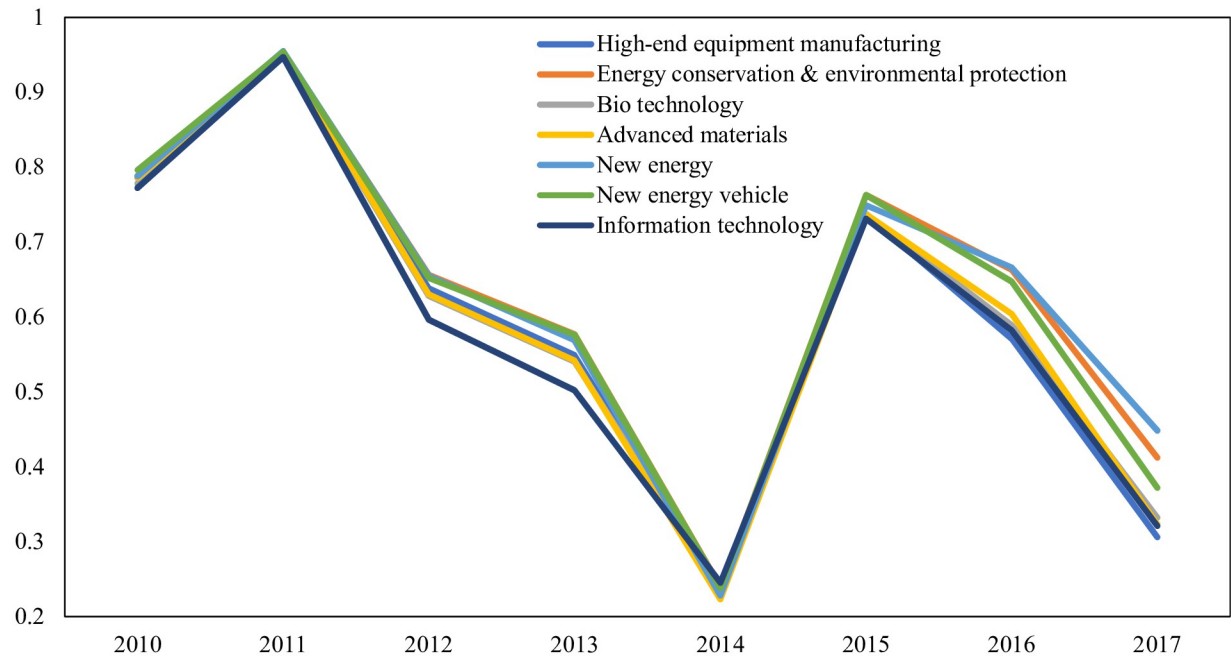

**Fig 1. Financing efficiency of China's strategic emerging industries.**

**Table 5. Variables and definition.**

| | Name | Description |
|---|---|---|
| **DV** | $Fineffi_{i,t}$ | Financing efficiency of firm $i$ at year $t$ |
| **IV** | $Finecol_{i,t-1}$ | Financial ecology of firm $i$'s city at year $t-1$ |
| **CV** | $TA_{i,t}$ | Total asset of firm $i$ at year $t$ |
| | $ATR_{i,t}$ | Asset-turnover ratio of firm $i$ at year $t$ |
| | $ALR_{i,t}$ | Asset-liability ratio of firm $i$ at year $t$ |
| | $FCFPS_{i,t}$ | Free cash flow per share of firm $i$ at year $t$ |
| | $Age_{i,t}$ | Firm age of firm $i$ at year $t$ |
| | $Year$ | Seven dummies to indicate eight years |
| | $Indt$ | Six dummies to indicate seven strategic emerging industries |
| | $Region$ | Two dummies to indicate eastern, central, and western regions |
| | $Board$ | Three dummies to indicate four types of China's market boards |

industries: high-end equipment manufacturing (HEM), energy conservation & environmental protection (ECEP), biotechnology (BT), advanced materials (AM), new energy (NE), new energy vehicle (NEV), and information technology (IT). Data for city financial ecology evaluation mainly comes from the "China City Statistical Yearbook (2009–2016)" issued by the National Bureau of Statistics of China and the statistical yearbook of each province. Securities market data for firm financing efficiency evaluation comes from the Wind database. After checking the data availability, this study collects a total of 32,104 pieces of data from 4,013 listed firms that are distributed in these strategic emerging industries.

Table 6 shows the distribution of firms in different strategic emerging industries and regions. Firms are mainly distributed in the HEM industry, followed by the NEV, BT, ECEP, IT, AM, and NE industries, reflecting different levels of China's strategic emerging industry development. Additionally, more than half of the firms are distributed in Eastern China with highly developed economy, highlighting the significant influence of economic factors on the agglomeration of strategic emerging industries.

Fig 2 shows the proportion of firms in different industries and regions. Overall, the industry distribution proportions of firms in different regions are roughly similar. Specifically, there are still subtle differences. The proportion of firms in the IT industry in Eastern China is significantly higher than that in Central China and Western China, while the proportion of firms in the HEM, ECEP, and BT industries in Western China is more balanced than that in Eastern China and Western China.

Table 7 shows the descriptive statistics of firm financing efficiency. The sample data can essentially be recognized as normal distribution since the absolute value of skewness is less than 3 and the absolute value of kurtosis is less than 10. The sample mean is 0.604 overall,

**Table 6. Firm distribution.**

| | Overall | HEM | ECEP | BT | AM | NE | NEV | IT |
|---|---|---|---|---|---|---|---|---|
| **Overall** | 4,013 | 1,254 | 559 | 656 | 226 | 186 | 673 | 510 |
| **Eastern China** | 2,880 | 933 | 365 | 413 | 151 | 143 | 482 | 422 |
| **Central China** | 526 | 173 | 84 | 108 | 31 | 22 | 78 | 42 |
| **Western China** | 607 | 148 | 110 | 135 | 44 | 21 | 113 | 46 |

Notes: The total number of firms is less than the sum of the number of firms in different industries because some firms are distributed in multiple industries.

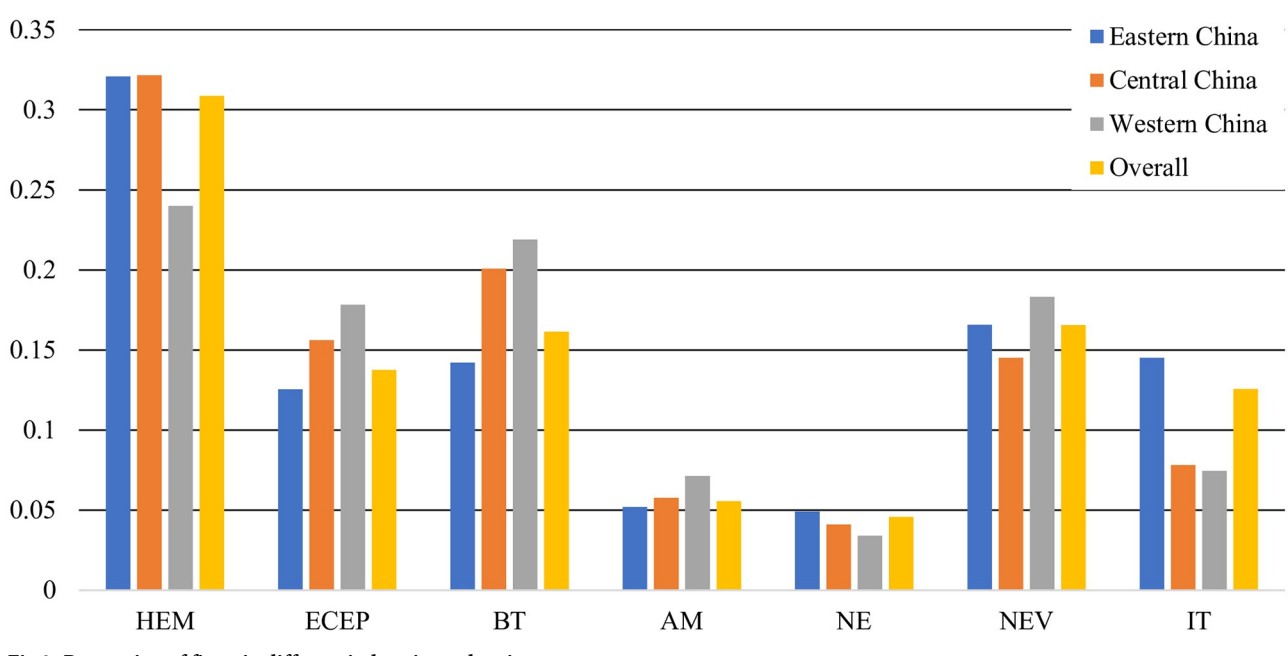

**Fig 2. Proportion of firms in different industries and regions.**

indicating that there is still potential for improvement in the firm's financing efficiency. In terms of industry, the ECEP and AM industries have the highest firm financing efficiency (0.632), while the IT industry has the lowest (0.587). In terms of region, Western China has the highest firm financing efficiency (0.608), while Central China has the lowest (0.598).

Table 8 show the descriptive statistics of city financial ecology. The sample data can essentially be recognized as normal distribution since the absolute value of skewness is less than 3 and the absolute value of kurtosis is less than 10. The sample mean is 0.345 overall, indicating that the city financial ecology urgently needs to be improved. In terms of industry, the IT industry has the highest city financial ecology (0.405), while the NEV industry has the lowest

**Table 7. Descriptive statistics of firm financing efficiency.**

|  | Min | Max | Mean | SD | Skewness | Kurtosis |
|---|---|---|---|---|---|---|
| **Overall** | 0.035 | 1.000 | 0.604 | 0.247 | -0.324 | -1.025 |
| **Industries** | | | | | | |
| **HEM** | 0.044 | 1.000 | 0.596 | 0.248 | -0.281 | -1.059 |
| **ECEP** | 0.077 | 1.000 | 0.632 | 0.241 | -0.467 | -0.900 |
| **BT** | 0.051 | 1.000 | 0.599 | 0.247 | -0.302 | -1.043 |
| **AM** | 0.114 | 1.000 | 0.632 | 0.241 | -0.507 | -0.794 |
| **NE** | 0.079 | 1.000 | 0.625 | 0.247 | -0.426 | -0.943 |
| **NEV** | 0.094 | 1.000 | 0.600 | 0.245 | -0.316 | -1.007 |
| **IT** | 0.035 | 1.000 | 0.587 | 0.249 | -0.208 | -1.101 |
| **Regions** | | | | | | |
| **Eastern China** | 0.035 | 1.000 | 0.604 | 0.247 | -0.319 | -1.029 |
| **Central China** | 0.077 | 1.000 | 0.598 | 0.247 | -0.314 | -1.037 |
| **Western China** | 0.093 | 1.000 | 0.608 | 0.246 | -0.360 | -0.989 |

**Table 8. Descriptive statistics of city financial ecology in different regions.**

|  | Min | Max | Mean | SD | Skewness | Kurtosis |
|---|---|---|---|---|---|---|
| **Overall** | 0.043 | 0.597 | 0.345 | 0.108 | 0.631 | -0.337 |
| **Industries** | | | | | | |
| **HEM** | 0.043 | 0.597 | 0.334 | 0.101 | 0.685 | -0.115 |
| **ECEP** | 0.043 | 0.597 | 0.355 | 0.108 | 0.602 | -0.358 |
| **BT** | 0.043 | 0.597 | 0.328 | 0.100 | 0.694 | 0.060 |
| **AM** | 0.043 | 0.597 | 0.346 | 0.107 | 0.613 | -0.299 |
| **NE** | 0.150 | 0.597 | 0.333 | 0.098 | 0.581 | -0.264 |
| **NEV** | 0.043 | 0.597 | 0.327 | 0.104 | 0.804 | 0.010 |
| **IT** | 0.043 | 0.597 | 0.405 | 0.117 | 0.141 | -1.137 |
| **Regions** | | | | | | |
| **Eastern China** | 0.129 | 0.597 | 0.364 | 0.114 | 0.442 | -0.827 |
| **Central China** | 0.111 | 0.425 | 0.276 | 0.064 | 0.218 | -1.034 |
| **Western China** | 0.043 | 0.435 | 0.313 | 0.066 | -0.448 | -0.203 |

(0.327). In terms of region, Eastern China has the highest firm financing efficiency (0.364), while Central China has the lowest (0.276).

 **Results of regressions.** Table 9 shows the results of regressions with industry-grouped samples. Overall, city financial ecology positively affects firm financing efficiency in all

**Table 9. Results of random-effect Tobit regressions on firm financing efficiency with industry-grouped samples.**

|  | Overall | HEM | ECEP | BT | AM | NE | NEV | IT |
|---|---|---|---|---|---|---|---|---|
| *Finecol* | 0.230** | 0.219*** | 0.204*** | 0.277*** | 0.262*** | 0.209*** | 0.249*** | 0.217*** |
|  | (0.013) | (0.024) | (0.034) | (0.034) | (0.541) | (0.067) | (0.032) | (0.034) |
| *TA* | -0.009*** | -0.011** | 0.008 | -0.021*** | 0.007 | 0.021** | -0.023*** | -0.019*** |
|  | (0.002) | (0.005) | (0.006) | (0.006) | (0.009) | (0.010) | (0.006) | (0.007) |
| *ATR* | 0.201*** | 0.390*** | 0.209*** | 0.156*** | 0.069 | 0.200** | 0.110** | 0.129** |
|  | (0.021) | (0.044) | (0.079) | (0.042) | (0.092) | (0.095) | (0.049) | (0.063) |
| *ALR* | 0.370*** | 0.331*** | 0.537*** | 0.340*** | 0.429*** | 0.412*** | 0.391*** | 0.256*** |
|  | (0.016) | (0.031) | (0.039) | (0.40) | (0.068) | (0.072) | (0.039) | (0.043) |
| *FCFPS* | -0.284*** | -0.303*** | -0.334*** | -0.271*** | -0.284*** | -0.288*** | -0.280*** | -0.223*** |
|  | (0.007) | (0.013) | (0.018) | (-0.17) | (0.029) | (0.032) | (0.017) | (-0.019) |
| *Age* | -0.139*** | -0.147*** | -0.141*** | -0.119*** | -0.155*** | -0.129*** | -0.152*** | -0.151*** |
|  | (0.004) | (0.006) | (0.009) | (0.007) | (0.017) | (0.017) | (0.010) | (0.010) |
| *Board dummies* | Included | Included | Included | Included | Included | Included | Included | Included |
| *Region dummies* | Included | Included | Included | Included | Included | Included | Included | Included |
| *Constant* | 1.242*** | 1.264*** | 1.088*** | 1.231*** | 1.231** | 0.942*** | 1.395*** | 0.907*** |
|  | (0.031) | (0.055) | (0.069) | (0.076) | (0.119) | (0.121) | (0.073) | (18.12) |
| Number of obs | 32,104 | 10,032 | 4,472 | 5,246 | 1,808 | 1,488 | 5,384 | 4,079 |
| Number of firms | 4,013 | 1,254 | 559 | 656 | 226 | 186 | 674 | 510 |
| Log likelihood | 1236.22 | 362.70 | 307.78 | 203.43 | 110.58 | 51.32 | 222.28 | 95.40 |
| Wald chi2 test | 3907.01*** | 1288.34*** | 635.44*** | 655.81*** | 217.02*** | 168.69*** | 617.59*** | 454.32*** |

Notes: Standard error in parentheses,

*** p<0.01,

** p<0.05,

* p<0.1

strategic emerging industries ($\beta = 0.230$, $p < 0.05$), aligning with previous findings [11, 16]. Three components make up the internal mechanism of financial ecology that supports the growth of strategic emerging industries: the first is financial scale support, which can quickly collect scattered capital from society to create a strong funding supply for strategic emerging industries, thereby meeting the funding needs for industrial development and growth; the second is financial structure support, which means that the more developed finance is, the more financing channels firms can obtain to reducing financing costs; the third is financial efficiency support, which means that varied and effective financial services may boost important growing sectors and increase the efficiency of resource allocation to meet capital demands more rapidly, affordably, and consistently [29]. Therefore, numerous city governments have implemented policies to improve their financial ecologies in light of the reliance of development of strategic emerging industries on it. For example, Beijing has introduced "Beijing's 14th Five Year Plan for the Development of the Financial Industry", which proposes to support the development of advanced manufacturing industry under the direction of the construction of a high-grade, precision and advanced economic structure, and guide financial institutions to increase their support for the two pillar industries (new generation information technology, and medicine & health), as well as the four characteristic advantageous industries (integrated circuits, intelligent connected vehicles, intelligent manufacturing and equipment, green energy and energy conservation & environmental protection); Shanghai has introduced "14th Five Year Plan for the Construction of Shanghai International Financial Center", which proposes to support industries (e.g. integrated circuits, biopharmaceuticals, and artificial intelligence) to accelerate development through capital markets such as the Science and Technology Innovation Board, and guide financial institutions to develop new products and services, increase the scale of medium- and long-term loans, and extend credit loans to provide high-quality financial support for key industries (e.g. electronic information, life and health, automobiles, high-end equipment, and advanced materials). Going beyond that, the impacting strength differs among industries. It is the strongest in BT industry ($\beta = 0.277$, $p < 0.01$), then AM ($\beta = 0.262$, $p < 0.01$), NEV ($\beta = 0.249$, $p < 0.01$), HEM ($\beta = 0.219$, $p < 0.01$), IT industry ($\beta = 0.217$, $p < 0.01$), NE ($\beta = 0.209$, $p < 0.01$) in sequence, and the weakest in ECEP ($\beta = 0.204$, $p < 0.01$). The findings suggest that different strategic emerging industries have differential market size, innovation ability and growth potentiality, leading to different dependence on external financial ecology for financing. We can divide these strategic emerging industries into three tiers, with each tier corresponding to an ideal financing model [30]: The first tier includes the three industries of BT, AM, and NEV, characterized by large asset bases, high equity levels, and propensity for profitability and innovation, where firms are suitable for equity financing, debt financing, and commercial credit financing. The second tier includes the two industries of HEM and IT, characterized by significant differentiation in firm development levels, where firms in mature stage are suitable for equity financing and debt financing similar to industries belonging to the first tier, while firms in early growth stage also require appropriate venture capital and government financial support. The third tier includes the two industries of NE and ECEP, characterized by small asset size, low net profit, and significant financing gaps for R&D and market expansion, where firms rely more on venture capital and fiscal funds.

Table 10 shows the results of regressions with region-grouped samples. Among all sample cities, there are 85 located in Eastern China, 88 in central China, and 115 in Western China. Overall, city financial ecology positively affects firm financing efficiency ($\beta = 0.230$, $p < 0.05$). However, the impacting strength also differs among regions. City financial ecology shows the strongest effect in Central China ($\beta = 0.669$, $p < 0.01$), the second strongest effect in Western China ($\beta = 0.376$, $p < 0.01$), and the weakest effect in Eastern China ($\beta = 0.190$, $p < 0.01$).

**Table 10. Results of random-effect Tobit regressions on firm financing efficiency with region-grouped samples.**

|  | Overall | Eastern China | Central China | Western China |
|---|---|---|---|---|
| *Finecol* | 0.230** | 0.185*** | 0.669*** | 0.416*** |
|  | (0.013) | (0.014) | (0.056) | (0.051) |
| *TA* | -0.009*** | -0.008*** | 0.002 | 0.004 |
|  | (0.002) | (0.003) | (0.007) | (0.006) |
| *ATR* | 0.201*** | 0.178*** | 0.175*** | 0.258*** |
|  | (0.021) | (0.024) | (0.064) | (0.072) |
| *ALR* | 0.370*** | 0.383*** | 0.369*** | 0.344*** |
|  | (0.016) | (0.019) | (0.046) | (0.038) |
| *FCFPS* | -0.284*** | -0.283*** | -0.288*** | -0.284*** |
|  | (0.007) | (0.008) | (0.019) | (-16.36) |
| *Age* | -0.139*** | -0.138*** | -0.153*** | -0.133*** |
|  | (0.004) | (0.004) | (0.009) | (0.009) |
| *Industry dummies* | Included | Included | Included | Included |
| *Region dummies* | Included | Included | Included | Included |
| *Constant* | 1.242*** | 1.218*** | 1.108*** | 1.114*** |
|  | (0.031) | (0.033) | (0.084) | (0.081) |
| Number of obs | 32,104 | 23,040 | 4,208 | 4,856 |
| Number of firms | 4,013 | 2,880 | 526 | 607 |
| Log likelihood | 1236.22 | 799.01 | 239.19 | 209.27 |
| Wald chi2 test | 3907.01*** | 2638.85*** | 696.50*** | 596.32*** |

Notes: Standard error in parentheses,

*** $p < 0.01$,

** $p < 0.05$,

* $p < 0.1$

Although the financing efficiency of China's strategic emerging industries in various regions is still not optimistic [10], the dominant factors that lead to weaker effect in Western China and Eastern China are different. For the western region, the dominant reason is the absolute shortage of financial resources. In 2022, total domestic and foreign currency deposits of financial institutions in China exceeded 264 trillion yuan (excluding Hong Kong, Macao, and Taiwan), with the western region accounting for less than 45 trillion yuan (17% of the total), while the eastern region exceeding 152 trillion yuan (57.6% of the total). Due to the significant financing constraints placed on strategic emerging industries as the result of limited financial resources, it is difficult for startups in the western region to increase their financing efficiency through rapid expansion. For the eastern region, the dominant reason is the redundancy of financial support [31]. Among them, direct financial support represented by the capital market and indirect financial support represented by banks have more redundancy, while equity investment support represented by venture capital has less redundancy. Therefore, the eastern region needs to optimize the financial ecosystem through innovation in financial products and financial systems, further achieving the rational allocation of financial resources.

## Conclusion and policy implication

The conclusions are as follows: (1) City financial ecology differs among regions. Overall, Eastern China is superior to Central China and Western China. Also, regions of Beijing-

Tianjin, the Yangtze River Delta, and the Pearl River Delta are better than other metropolitan areas. Beijing, Shanghai, Shenzhen, Tianjin and Guangzhou rank top five cities. (2) The financing efficiency of China's strategic emerging firms is not optimistic, showing a fluctuating downward trend. It is due to the prudent fiscal and monetary policies. Since 2011, the central government of China has implemented conservative fiscal and monetary policies. Despite a loose policy during 2014 and 2015, the central government soon began new supply-side reform and cut-overcapacity policies. (3) City financial ecology positively affects strategic emerging firms' financing efficiency overall, but its impact differs among industries and regions. Specifically, city financing ecology plays the most substantial role in the BT industry while the weakest in the IT industry. It works the most in Central China, Western China, and Eastern China.

Considering the unsatisfied status quo of strategic emerging firms' financing and the essential role of cities' financial ecology, the governments, especially those in central China, urgently need to attach great importance to financial ecology and optimize it through better policymaking. Guidelines from financial ecology growth, soil, and air perspectives are as follows: (1) The government should improve the multi-tiered financial market system. According the evaluation result of the city's financial ecology, there is a regional imbalance in China's financial ecology. Therefore, the government should speed up the financial ecosystem's self-development by creating a multi-level financial service system and encouraging financial innovation in order to achieve diversified financial competition and financial products and to fully exploit the role of finance in fostering the development of strategic emerging firms. The government should also actively promote foreign investment and international financial institutions' establishment of branches in China, as well as actively support domestic and foreign financial institutions' migration to regions with weak financial ecology. In addition, the government should organize multi-level, multi-channel, and multi-form "government-bank-firm" project funding docking activities to establish a platform channel for direct communication between banks and firms, which can improve efficient docking between important projects and financial institutions. (2) The government should provide differentiated support for the development of different industries. According to the result of mixed-effect Tobin panel regression, the impact of the financial ecology on financing efficiency varies industries and regions. Therefore, the government should design support policies based on the peculiarities of various sectors to accelerate regional economic development by transforming the economic growth model and support strategic emerging industries by optimizing and upgrading the industrial structure. For example, actively introducing and enhancing laws and regulations with industrial nature in terms of industrial development, financial support, etc., and offering various degrees of tilt based on various industries in terms of tax incentives, patent protection, business guidance, etc. (3) The government should encourage firms to continually enhance internal governance. According to the regression result of the control variables, the capability to manage and utilize internal assets of a firm has some bearing on financing efficiency, and the issue of financing challenges cannot be fully resolved by relying exclusively on improving the financial ecology and the assistance of outside financial institutions. Therefore, the government should promote the subjective initiative of firms, help them consistently overcome their own and external limitations, improve their capacity for innovation, and actively look for favorable development opportunities, thereby increasing their profitability and own funds, as well as the capability and effectiveness of internal financing. The government should also further improve the credit, cultural, and security environment required for firms to enhance their governance by establishing an information-sharing mechanism and constructing a social credit system.

## Limitations and future works

There are some limitations in this study: (1) indexes such as legal environment, institutional environment, and enterprise satisfaction are not included in the evaluation index system of city financial ecology due to data availability; (2) the sample consists of listed companies in China, which cannot fully represent the full picture of China's strategic emerging industries. Therefore, we will try to quantify the aforementioned indexes and broaden the sample size in future research in order to further examine the impact of city financial ecology on firm financing efficiency.

## Supporting information

**S1 Dataset.**
(XLSX)

## Author Contributions

**Data curation:** Xing Li.

**Methodology:** Yuchen Gao.

**Software:** Dong Wang.

**Writing – original draft:** Hanbo Zhang, Guiyang Zhang.

**Writing – review & editing:** Yong Qi.

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
