## [Decision Letter · Decision Letter 0]

23 Mar 2023

PONE-D-23-01009The impact of city financial ecology on firm financing efficiency: Evidence from Chinese strategic emerging industriesPLOS ONE

Dear Dr. Qi,

Thank you for submitting your manuscript to PLOS ONE. After careful consideration, we feel that it has merit but does not fully meet PLOS ONE’s publication criteria as it currently stands. Therefore, we invite you to submit a revised version of the manuscript that addresses the points raised during the review process.

We look forward to receiving your revised manuscript.

Kind regards,

Wajid Khan

Academic Editor

PLOS ONE

Journal Requirements:

Reviewers' comments:

Reviewer's Responses to Questions

**Comments to the Author**

1. Is the manuscript technically sound, and do the data support the conclusions?

Reviewer #1: Yes

Reviewer #2: Yes

2. Has the statistical analysis been performed appropriately and rigorously? 

Reviewer #1: Yes

Reviewer #2: Yes

3. Have the authors made all data underlying the findings in their manuscript fully available?

Reviewer #1: No

Reviewer #2: Yes

4. Is the manuscript presented in an intelligible fashion and written in standard English?

Reviewer #1: Yes

Reviewer #2: Yes

5. Review Comments to the Author

Reviewer #1: I have reviewed the manuscript titled “The impact of city financial ecology on firm financing efficiency: Evidence from Chinese strategic emerging industries”

Upon thorough observations I came across the following discrepancies in the manuscript and the authors should incorporate the following suggestions with due consideration.

1) Policy Implication statement is missing in the abstract.

2) Highlight whether the proposed econometric approach is best one matching with the economic question addressed in this paper. Try to put out this issue in the introduction.

3) Please explicitly elaborate on how this research work contributes to the extant literature or what facets make it novel.

4) Highlight the importance of financial institutions in uplifting the economy as such variables propels economic growth via difference supply and demand channels. The following articles may help them in this regard.

https://doi.org/10.1002/ijfe.2115

https://doi.org/10.1016/j.physa.2019.124106

5) Descriptive statistics is missing. Please report it.

6) The discussion section weak. Please back up your work with the existing pertinent literature.

7) The conclusion section should be improved based on the study's findings. The authors also need to mention the study limitations and future research directions.

Reviewer #2: Article is well prepared and formatted accordingly. tables and equations are formatted according to the guidelines. However, there is minor suggestion to recheck headings and text formatting and spacing following the tables in the manuscript.

6. PLOS authors have the option to publish the peer review history of their article (what does this mean?). If published, this will include your full peer review and any attached files.

Reviewer #1: **Yes: **Assad Ullah

Reviewer #2: No

---

## [Author Response · Author response to Decision Letter 0]

7 Jun 2023

Rebuttal letter

Response to the Academic Editor

Thank you very much for taking the time to consider my manuscript. I have tried to improve the manuscript as requested.

Kind regards,

Yong Qi

Journal Requirements:

I have revised my manuscript to meet PLOS ONE's style requirements.

I have rechecked the ‘Funding Information’ and confirm that the grant numbers are correct.

I have uploaded the data set as “S1_Dataset.xlsx” and marked it in the section "Supporting information".

I have removed the figure and presented the results in table format instead. Please see “Table 4. Score distribution of city financial ecology”.

I added 5 references in my revised manuscript as blow:

1. Ullah A, Zhao X, Kamal M A, Riaz A, Zheng B. Exploring asymmetric relationship between Islamic banking development and economic growth in Pakistan: Fresh evidence from a non‐linear ARDL approach. International Journal of Finance & Economics, 2021; 26(4): 6168-6187.

2. Ullah A, Zhao X, Kamal M A, Zheng J. Modeling the relationship between military spending and stock market development (a) symmetrically in China: An empirical analysis via the NARDL approach. Physica A: Statistical Mechanics and its Applications, 2020; 554: 124106.

3. Ren Z. Research on the efficiency of financial support for strategic emerging industries: Evidence from listed companies in seven major industries. Communication of Finance and Accounting, 2021; (4): 160-163.

4. Hu J. Study on heterogeneity and financing pattern matching of strategic emerging industries ——Panel data of 120 listed companies in strategic emerging industries. Journal of Social Sciences, 2020; (4):44-57.

5. Ma J, Wang J. On efficiency of financial support for strategic emerging industries ——Based on empirical analysis of the Yangtze River. Forum on Science and Technology in China, 2019; (10): 52-58.

Also, I have updated my reference list to ensure that it is complete and correct.

Response to Reviewers

Reviewer #1

Thank you very much for your valuable comments. I have placed my clariﬁcations and declared the corresponding changes to the manuscript after each of your points below.

I have reviewed the manuscript titled “The impact of city financial ecology on firm financing efficiency: Evidence from Chinese strategic emerging industries”

Upon thorough observations I came across the following discrepancies in the manuscript and the authors should incorporate the following suggestions with due consideration.

1) Policy Implication statement is missing in the abstract.

I have added policy implication statement to the abstract in my revised manuscript.

2) Highlight whether the proposed econometric approach is best one matching with the economic question addressed in this paper. Try to put out this issue in the introduction.

I have done this work in my revised manuscript, and put the contents in the section “Methodology”.

3) Please explicitly elaborate on how this research work contributes to the extant literature or what facets make it novel.

I have done this work in my revised manuscript. Please see the section “Introduction”.

4) Highlight the importance of financial institutions in uplifting the economy as such variables propels economic growth via difference supply and demand channels. The following articles may help them in this regard.

https://doi.org/10.1002/ijfe.2115

https://doi.org/10.1016/j.physa.2019.124106

The articles you recommended have greatly inspired me, and I have cited them in the section "Theoretical analysis " to further strengthen the theoretical support for the classification of "financial ecology growth" by highlighting the importance of financial institutions in uplifting the economy.

5) Descriptive statistics is missing. Please report it.

I have done this work in my revised manuscript. Please see the section “Results and discussion”.

6) The discussion section weak. Please back up your work with the existing pertinent literature.

I have strengthened the discussion section in my revised manuscript.

7) The conclusion section should be improved based on the study's findings. The authors also need to mention the study limitations and future research directions.

I have done these works in my revised manuscript. Please see the section “Conclusion and policy implication” and “Limitations and future works”.

Reviewer #2

Thank you very much for the time invested and for the valuable comments.

Article is well prepared and formatted accordingly. tables and equations are formatted according to the guidelines. However, there is minor suggestion to recheck headings and text formatting and spacing following the tables in the manuscript.

I have rechecked and revised them to meet PLOS ONE's style requirements.

---

## [Editor Report · Decision Letter 1]

22 Jun 2023

The impact of city financial ecology on firm financing efficiency: Evidence from China's strategic emerging industries

PONE-D-23-01009R1

Dear Dr. Yong Qi,

We’re pleased to inform you that your manuscript has been judged scientifically suitable for publication and will be formally accepted for publication once it meets all outstanding technical requirements.

Kind regards,

Wajid Khan

Academic Editor

PLOS ONE

---

## [Editor Report · Acceptance letter]

26 Jul 2023

PONE-D-23-01009R1 

The impact of city financial ecology on firm financing efficiency: Evidence from China’s strategic emerging industries 

Dear Dr. Qi:

I'm pleased to inform you that your manuscript has been deemed suitable for publication in PLOS ONE. Congratulations! Your manuscript is now with our production department. 

Kind regards, 

on behalf of

Dr. Wajid Khan 

Academic Editor

PLOS ONE